# Preparation, Corrosion Resistance, and Electrochemical Properties of MnO$_2$/TiO$_2$ Coating on Porous Titanium

Xiaomin Wang, Jing Pan, Qing Li, Xiaojun Dong, Lei Shi * and Sujie Chang *

School of Materials Science and Engineering, Shandong Jianzhu University, Fengming Road, Jinan 250101, China
* Correspondence: slcqj@sdjzu.edu.cn (L.S.); 13771@sdjzu.edu.cn (S.C.); Tel.: +86-1300-659-4941 (L.S.);
+86-1509-879-5090 (S.C.)

**Abstract:** In this work, MnO$_2$/TiO$_2$ coating on metallic porous titanium was prepared through a hydrothermal-based chemical method, followed by a chemical precipitation reaction of KMnO$_4$ and MnSO$_4$ aqueous solutions. The surface of the MnO$_2$/TiO$_2$/Ti was uniform and compact, with a high load capacity. The corrosion resistance and electrochemical properties of the MnO$_2$/TiO$_2$/Ti coating were investigated in comparison with those of pure Ti and TiO$_2$ coatings. Cyclic voltammetry and constant current charge–discharge measurements showed that the MnO$_2$/TiO$_2$/Ti electrode presented good electrochemical performance. The MnO$_2$/TiO$_2$/Ti electrode had the highest capacitor performance compared to the other electrodes, and the nano-MnO$_2$ coating significantly decreased the corrosion current densities. The nano-MnO$_2$ coating exhibited excellent anti-corrosion properties at room temperature and better capacitance performance compared with pure Ti and TiO$_2$ coatings. After surface modification, TiO$_2$/Ti-coated MnO$_2$ had better electrochemical behavior and significantly improved corrosion resistance than the TiO$_2$/Ti nanocomposites. Its specific capacitance reached 314 F/g, which was 3.5 times that of the TiO$_2$/Ti electrode material.

**Keywords:** MnO$_2$/TiO$_2$; nano-coating; capacitance performance; corrosion resistance

## 1. Introduction

With the advent of the 5G era, environmental pollution and the clean energy crisis have become serious social problems in the process of economic and social progress, and increasingly more research is focused on the development of new energy storage systems [1]. As an important energy storage system, supercapacitors have been widely studied for their high power density and fast charge and discharge [2–5]. Supercapacitors are generally divided into double-layer capacitors (EDLCs) and pseudocapacitors. EDLCs store energy electrostatically through surface ion adsorption/desorption at the electrode/electrolyte interface, while pseudocapacitors utilize fast and reversible superficial Faradaic reactions between electrolyte ions and electroactive materials [6–8]. In general, the value of an EDLCs is much less than that of a psedocapacitor [9]. Manganese dioxide (MnO$_2$) is a promising pseudocapacitive material that has attracted widespread attention for use in many electroactive materials due to its low cost, low natural abundance, wide electrochemical potential window, and high theoretical capacitance value (1370 F/g) [10,11]. However, the inherently poor conductivity and easy dissolution of manganese dioxide hinder the realization of its high electrochemical properties. In contrast, TiO$_2$ has a higher electrical conductivity and electrochemical stability compared with MnO$_2$ [12].

Combining manganese dioxide with other materials (such as titanium dioxide [13], zinc oxide [14], and diferric trioxide [15]) to improve the conductivity and stability of the electrode material and leverage its advantages [13,16–19] may be an effective way to overcome these problems. Compared to manganese dioxide, titanium dioxide has higher electrical conductivity and electrochemical stability [20]. The use of binary metal oxide nanocomposite electrode materials has received widespread attention due to their special

physical properties and potential applications [21]. Titanium dioxide is one of the most studied non-silicon mesoporous metal oxides due to its wide applications in photocatalysis, solar cells, chemical sensors, and bioanalytical chemistry [22–25]. Furthermore, Kaseem and co-workers reported the importance of $TiO_2$ coating in electrochemical and biomedical applications based on plasma electrolytic oxidation or micro-arc oxidation coating treatment [26–29]. However, due to the high price of titanium raw materials and the difficulty of manufacturing, these are not often used in daily life. Titanium matrix composites have the advantages of being lightweight with high strength, good corrosion resistance, and good biocompatibility. $TiO_2$ nanomaterials have a high degree of order and a large specific surface area, so that the transmission path between ions and other electrons is reduced. This reduces the disadvantage of its poor conductivity. Furthermore, $MnO_2$ can be deposited on its surface, thereby increasing the capacitance. In addition, the electrochemical performance of $MnO_2$ is limited by the electronic conductivity of densely packed matter without a porous structure [30].

In this paper, we successfully prepared a novel $MnO_2/TiO_2/Ti$ sandwich nanostructure by a chemical co-precipitation reaction of potassium permanganate ($KMnO_4$) and manganese sulfate ($MnSO_4$) with an alternating immersion method. First, we placed purchased commercial foam titanium into a reactor for a hydrothermal reaction after anhydrous ethanol treatment, followed by annealing the titanium at 600 °C in a muffle furnace, and finally used a $KMnO_4$, $H_2O$, and $MnSO_4$ for alternating soaking treatments to obtain a nanocomposite electrode material with good conductivity through a further redox deposition reaction. The prepared $MnO_2/TiO_2/Ti$ nanocomposites exhibited excellent corrosion resistance, specific capacitance, and cycle stability.

## 2. Experimental

### 2.1. Metallic Porous Titanium Processing

In this study, metallic porous titanium (with a thickness of about 1.0–1.20 mm, Ti purity > 99.9 wt%, Suzhou Terry Foam Metal Factory, China) was used as the raw material. The porous titanium was cut into small rectangular pieces with the specifications of width × length × thickness = 20 mm × 40 mm × 1 mm. The porous Ti plates were then ultrasonically cleaned with acetone, ethanol, and ultrapurified water for 15 min sequentially and then dried at room temperature [31].

### 2.2. Preparation of TiO_2/Ti Nanomaterial

The $TiO_2/Ti$ nanomaterial was prepared via a facile alkaline hydrothermal method, which was derived from the method proposed for constructing a porous titanate layer on the surface of a titanium foil [31]. First, 20 mL of aqueous NaOH (98.0%, Jiangyin Runma Electronic Materials Co., Ltd., Jiangyin, China) solution (10 mol/L) was prepared. The aqueous NaOH solution was cooled to room temperature, and the cooled NaOH solution and foaming titanium were placed in a reaction kettle successively. The reaction kettle was placed in a vacuum-drying oven for hydrothermal treatment at 110 °C for 24 h. The kettle was removed under ambient pressure, cooled to room temperature, and the samples were removed with clean forceps. The samples were washed with deionized water to thoroughly remove the porous Ti to obtain the $Na_2Ti_3O_7$ nanomaterials. The nanomaterials were then immersed in 0.1 M HCl (36.0% to 38.0%, Chongqing Chuandong Chemical Group Co., Ltd., Chongqing, China) aqueous solution for 24 h at room temperature. The acid-treated $Na_2Ti_3O_7$ nanomaterials were thoroughly washed with deionized water to obtain the $H_2Ti_3O_7$ nanomaterials [32]. The crucible was soaked in nitric acid (60%, Chongqing Chuandong Chemical Group Co., Ltd., China) for 15 min, and the acid-treated crucible was neutralized with NaOH solution. The treated crucible was cleaned with deionized water until the pH of the cleaning solution was neutral. The $H_2Ti_3O_7$ nanomaterials were placed in the treated crucible and heated in a muffle furnace at 600 °C for 2 h for annealing to finally obtain the $TiO_2$ nanoelectrode materials.

### 2.3. Preparation of $MnO_2/TiO_2/Ti$ nanocomposite Electrode Materials

The $MnO_2/TiO_2/Ti$ nanocomposite electrode material was prepared using an alternating immersion method. The previously annealed $TiO_2$ electrode nanomaterials were immersed in 0.15 mol/L manganese sulfate (99.0%, Tianjin Dengfeng Chemical Reagent Factory, Tianjin, China) solution for 30 s, then removed and immersed in deionized water for 30 s, then immersed in 0.1 mol/L potassium permanganate (analytically pure, Group Chemical Reagent Co., Ltd., Shanghai, China) solution for 30 s, and finally immersed in deionized water for 30 s to complete an alternating immersion cycle. The operation was repeated six times. At the end of the complete process, the samples were thoroughly cleaned with deionized water by soaking in the water for 15 min and then rinsing with running deionized water. The samples were dried in a constant-temperature drying oven for 2 h at 60 °C. At the end of the process, the samples were sealed and stored for future use.

### 2.4. Material Property Characterization and Electrochemical Testing

The crystalline structures of the products were characterized by X-ray diffraction (XRD) in a Bruker D8 Advance powder X-ray diffractometer with Cu Kα radiation ($\lambda$ = 0.15406 nm). Raman scattering has been applied previously to the structural characterization of manganese dioxides; therefore, a Laba RAM HR Evolution UV Raman spectrometer (Horiba, France) was used to analyze the Raman spectra of the experimental samples. The data point acquisition time was 30 s. The excitation was performed under a diode laser with an excitation wavelength of 532 nm and a laser power of 9.1 mW, and the spectral region was 50–900 nm. The morphologies of the prepared electrode materials were characterized using a scanning electron microscope (SEM). The SEM images were taken using an SU3800 microscope. The properties of the prepared samples were tested using Tafel curves, cyclic voltammetry (CV), and electrochemical impedance spectroscopy (EIS) using a CHI760E electrochemical workstation. The tests were performed using a standard three-electrode system with titanium foam, $TiO_2/Ti$, and $MnO_2/TiO_2/Ti$ as the working electrodes, a platinum electrode as the auxiliary electrode, and a saturated glycogen electrode as the reference electrode. The electrolyte system of the test sample was a 0.5 mol/L KOH solution, the scanning potential was from −0.2 to 0.7 V, the scanning speed was 10 mV/s, and the number of scanning turns was 10.

## 3. Results and Discussion

The crystal phases of the commercialized Ti and the structure of the prepared $TiO_2/Ti$ and $MnO_2/TiO_2/Ti$ nanocomposite membrane materials were characterized by XRD. As can be seen from Figure 1, porous titanium and standard titanium metal (PDF#44-1924) corresponded very well, with no stray peaks. Except for the peaks corresponding to metal titanium, the diffraction peaks of $TiO_2/Ti$ corresponded well to $TiO_2$ anatase phase peaks (PDF#21-1272). The strongest peaks at 2θ = 25.4°, 37.8°, 48.0°, 53.9°, and 55.1° corresponded to the (101), (004), (200), (105), and (211) surfaces of the anatase phase, respectively [33]. The diffraction peak of $MnO_2$ was not clearly evident in the XRD pattern, and further characterization was required to determine the presence of $MnO_2$.

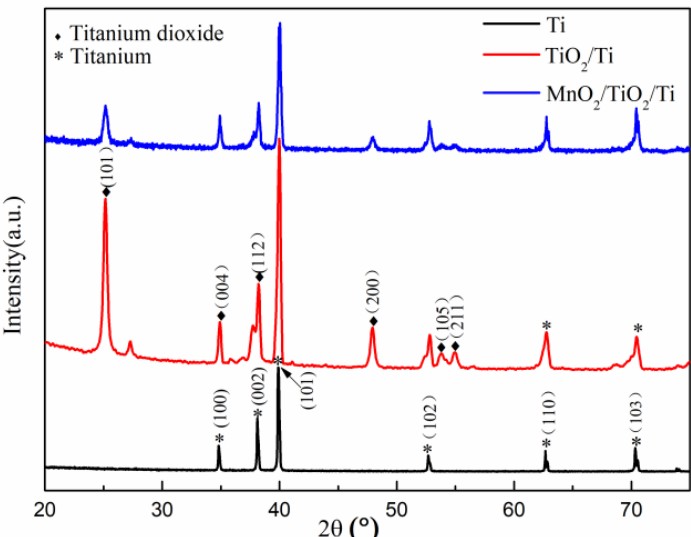

**Figure 1.** X-ray diffraction patterns of commercialized metal porous Ti, surface-modified $TiO_2$/Ti, and $MnO_2$/$TiO_2$/Ti nanocomposites.

Raman spectroscopy was performed on the metal porous Ti and the prepared $TiO_2$/Ti and $MnO_2$/$TiO_2$/Ti nanocomposite samples. The Raman peaks shown in the Ti and $TiO_2$/Ti curves of Figure 2 were found at 144 cm$^{-1}$, 196 cm$^{-1}$, 394 cm$^{-1}$, 515 cm$^{-1}$, 519 cm$^{-1}$, and 636 cm$^{-1}$ and confirmed that the surface-modified $TiO_2$/Ti nanocomposite conformed to the crystal form of anatase [32], consistent with the diffraction results of the X-ray analysis in Figure 1. In addition to the Raman peaks of the anatase phase of $TiO_2$, the Raman peak of $MnO_2$ can be observed at 575 cm$^{-1}$, which can be seen in the $MnO_2$/$TiO_2$/Ti curve. At 575 cm$^{-1}$, the peak strength of the $MnO_2$/$TiO_2$/Ti curve was significantly increased compared with the 636 cm$^{-1}$ Raman peak of the $TiO_2$/Ti curve, which may contain both $MnO_2$ and $TiO_2$. This suggested that the substance in the $MnO_2$/$TiO_2$/Ti nanocomposite sample prepared by alternating immersion deposition was $MnO_2$ particles.

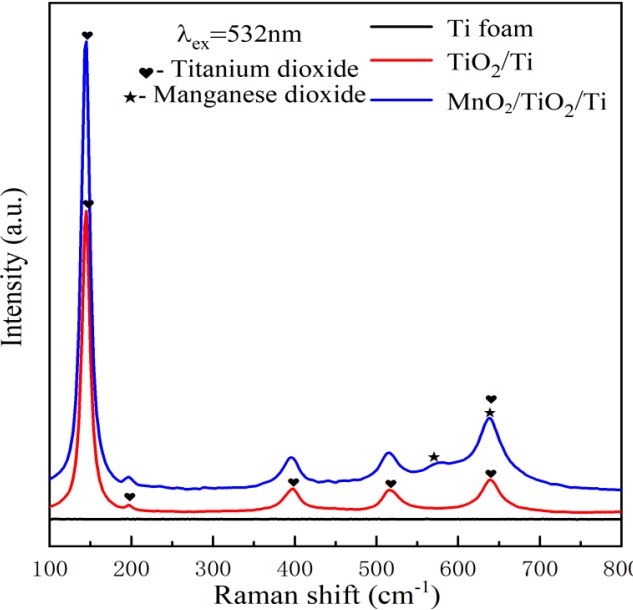

**Figure 2.** Raman spectroscopy test results of commercialized metal porous Ti, surface−modified $TiO_2$/Ti, and $MnO_2$/$TiO_2$/Ti nanocomposites.

The microstructures of the metallic porous Ti, surface-modified $TiO_2$/Ti, and $MnO_2$/$TiO_2$/Ti nanocomposites were elucidated by scanning electron microscopy. The results are shown in Figure 3. It can be clearly seen that the internal structure of the metal titanium foam was porous, its surface was uneven (Figure 3a), and there were micropores with irregular shapes and different sizes (Figure 3b). The microstructure of the $TiO_2$/Ti prepared by the alkaline hydrothermal method is shown in Figure 3c,d. Here, $H_2Ti_3O_7$ was annealed, and local chemical reactions and dehydration processes occurred to generate an anatase-type $TiO_2$ [34]. The surface of the $TiO_2$/Ti electrode material was more flattened than the surface of the foam titanium, and small $TiO_2$ nanocrystals formed, which clustered to form a 3D nanowires network (Figure 3d), providing spaces for the subsequent loading of the $MnO_2$ nano-coating. Moreover, the surface structure of the $TiO_2$ was conducive to the transfer of ions in the redox Faraday reaction [35]. Often, different synthesis methods affect the forming structure of the material, and the growth mechanism of the crystals in the solution is actually quite complex and therefore not fully explained. With regard to the hydrothermal process, classical nucleation, orientation aggregation, and Ostwald ripening have been proposed [36,37]. Here, $KMnO_4$ and $MnSO_4$ were uniformly deposited on the $TiO_2$/Ti formed by a co-proportionation reaction. With Ostwald maturation, $MnO_2$ would be converted into small nanoparticles [38], and Figure 3e,f is consistent with the findings above. The uniformity of the ultra-thin manganese dioxide overlaid on the porous nanoelectrode enables the rapid transport of electrolyte ions and provides a continuous path for charge transfer [39]. The surface of the coating was uniform and compact, with a high load capacity. In summary, a nano-coating structure based on porous Ti provides good surface energy, which is conducive to the higher circulation performance of the electrode material.

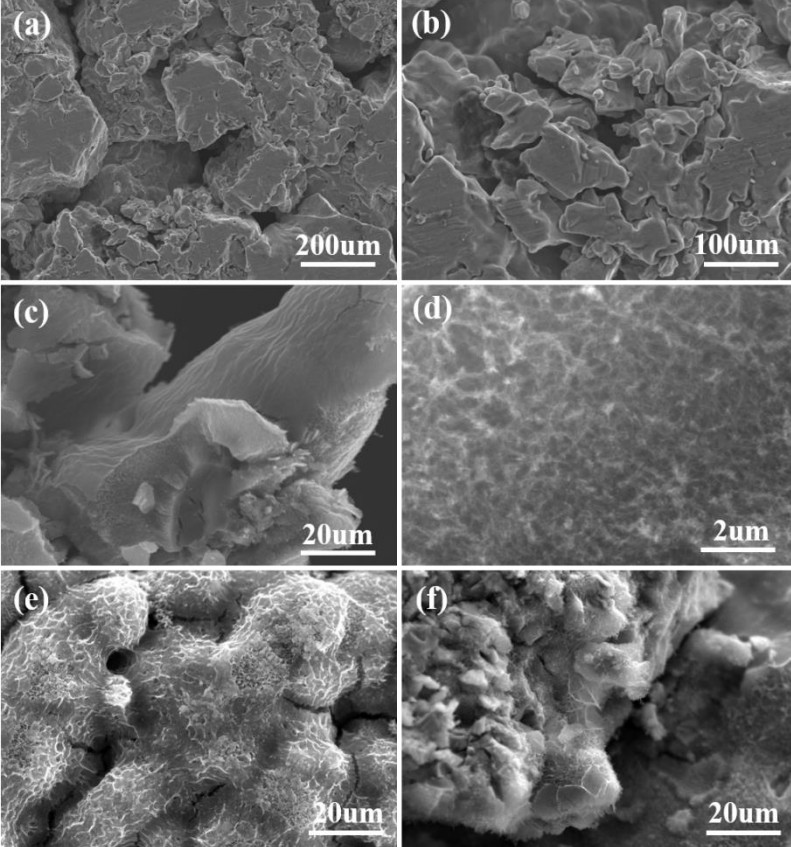

**Figure 3.** Scanning electron microscopy images of different structures of metal porous Ti (**a**,**b**), $TiO_2$/Ti (**c**,**d**), and $MnO_2$/$TiO_2$/Ti nanocomposites (**e**,**f**).

The Tafel polarization curve is used to compare the relationship between the corrosion potential of the coating and the corrosion current density, which allows the corrosion resistance of the electrode material to be explored. Xavier et al. verified that the incorporation of $MnO_2$ nanoparticles reduced the corrosion rate [40]. The presence of $MnO_2$ nanoparticles helps to clog pores and cracks in the coating, hindering the initiation of corrosion at the coating/metal interface. The Tafel curves of the Ti, $TiO_2$/Ti, and $MnO_2$/$TiO_2$/Ti electrodes are shown in Figure 4 and Table 1, and it can be seen that the corrosion current density (Icorr) of the surface-modified $MnO_2$/$TiO_2$/Ti electrode was significantly reduced compared with that of the Ti and $TiO_2$/Ti electrodes. In the presence of $MnO_2$, the self-corrosion potential (Ecorr) value shifted in the positive direction, which indicated that $MnO_2$/$TiO_2$/Ti significantly delayed the dissolution process of the electrode and improved its anti-corrosion performance [41]. This is consistent with the above conclusions.

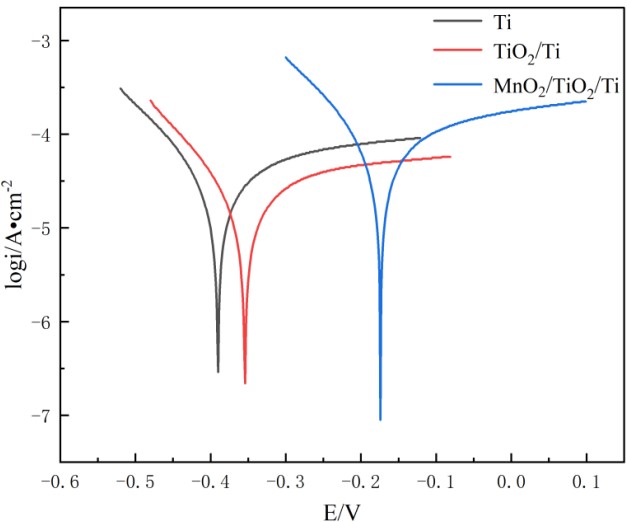

**Figure 4.** Tafel test results of metal porous Ti, surface−modified $TiO_2$/Ti, and $MnO_2$/$TiO_2$/Ti nanocomposites.

**Table 1.** Results of the Tafel texts of the porous Ti, surface-modified $TiO_2$/Ti, and $MnO_2$/$TiO_2$/Ti nano composites.

| Sample | Tafel Polarization | | | |
|---|---|---|---|---|
| | Icorr (uA) | Ecorr (mV) | Cathodic Tafel Slope (V/Decade) | Anodic Tafel Slope (V/Decade) |
| Ti | 36.08 | −320 | −8.221 | 3.019 |
| $TiO_2$/Ti | 23.93 | −260 | −9.633 | 2.873 |
| $MnO_2$/$TiO_2$/Ti | 80.41 | −100 | −8.224 | 3.091 |

To better understand the electrochemical properties of the electrode material, an electrochemical impedance analysis was performed. EIS measurements were made in the frequency range of 1 Hz–100 kHz, and the resulting Ti, $TiO_2$/Ti, and $MnO_2$/$TiO_2$/Ti Nyquist curves and Bode plots are shown in Figure 5. The Nyquist diagram (Figure 5a) consisted of a low-frequency slash and a semicircular arc phase of a medium and high frequency, the semicircular arc of the high frequency region represented the charge transfer resistance, and the Z′ and Z″ at the interface of the electrolyte and electrode were the real and imaginary parts of the impedance, respectively. The results obtained by the Bode plots (Figure 5b) were consistent with the Nyquist diagram. The equivalent circuit diagram is shown in Figure 5 [41], where Rs is the solution resistance, Rct is the interfacial charge transfer resistor, where the size of the Rct depends on the electrode area that the electrolyte can access and the conductivity of the electrode, the smaller the Rct indicates the faster the

electron transfer and ion diffusion, and CPE is the constant phase angle. The capacitance values for Ti, $TiO_2/Ti$, and $MnO_2/TiO_2/Ti$ were 1.556 F, 1.315 F, and 1.573 F, respectively, and the results showed that the capacitance values of $MnO_2/TiO_2/Ti$ were significantly higher than those of both of the former. As can be seen from Table 2 and Figure 5, the Rs values (15.34 $\Omega cm^2$) of the $MnO_2/TiO_2/Ti$ electrodes were smaller than those of surface-modified $TiO_2/Ti$ (64.99 $\Omega cm^2$) and Ti (98.02 $\Omega cm^2$) nanocomposites, while the charge transfer resistance (4.20 $\Omega cm^2$) of the $MnO_2/RGO$ electrodes was also much smaller than that of $TiO_2/Ti$ (37.94 $\Omega cm^2$) and porous Ti (92.11 $\Omega cm^2$). In the low-frequency range, the electrode reaction kinetics were mainly controlled by diffusion (Warburg) [42], and the slope of the Nyquist plot curve of the electrode was increasing, indicating rapid ion diffusion and good capacitor performance [43–45].

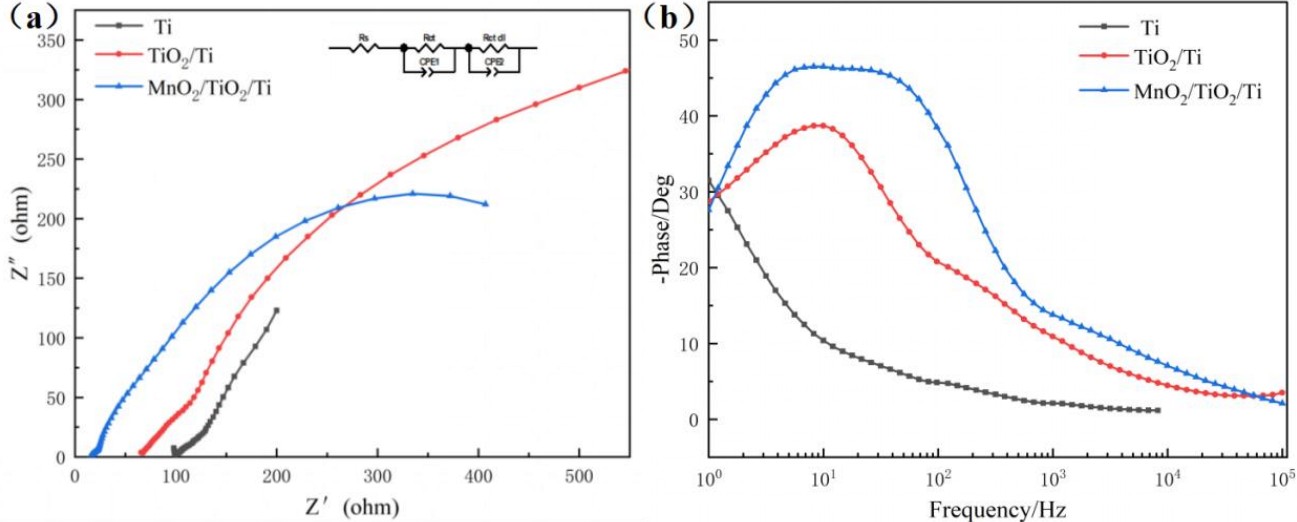

**Figure 5.** (**a**) Nyquist diagram and (**b**) Bode plots of metal porous Ti, surface-modified $TiO_2/Ti$ and $MnO_2/TiO_2/Ti$ nanocomposites and equivalent circuits.

**Table 2.** Results of the EIS fitting texts of the porous Ti, surface-modified $TiO_2/Ti$, and $MnO_2/TiO_2/Ti$ nano composites.

| Sample | Rs ($\Omega cm^2$) | Rct ($\Omega cm^2$) | CPE1-T | CPE-P | Rct dl ($\Omega cm^2$) | CPE2-T | CPE2-P |
|---|---|---|---|---|---|---|---|
| Ti | 98.02 | 92.11 | $2.7 \times 10^{-3}$ | 0.453 | 194.4 | $1.1 \times 10^{-3}$ | 1.10 |
| $TiO_2/Ti$ | 64.99 | 37.94 | $3.0 \times 10^{-4}$ | 0.580 | 986.1 | $2.2 \times 10^{-4}$ | 0.74 |
| $MnO_2/TiO_2/Ti$ | 15.34 | 4.20 | $3.17 \times 10^{-5}$ | 0.853 | 691.7 | $3.5 \times 10^{-4}$ | 0.72 |

The electrochemical properties of the electrode material were further studied by performing CV tests. Figure 6 shows the test results of commercialized metal foam Ti, surface-modified $TiO_2/Ti$, and $MnO_2/TiO_2/Ti$ nanocomposites in a KOH solution with an electrolyte system of 0.5 mol/L. The scanning potential was from −0.2 to 0.7 V, and the scanning speed was 10 mV/s. As shown in Figure 6a, all the CV curves in the voltage range had a symmetrical shape. Each of the three samples were analyzed by taking ten turns. Figure 6b presents the graph of the test results for one cycle of each sample, which showed that the $MnO_2/TiO_2/Ti$ curve was shaped relative to a rectangle with a small redox peak with $TiO_2/Ti$ (Faraday reaction) [13], indicating the pseudo-capacitance behavior of the electrode. Compared with the $TiO_2/Ti$ nanocomposite, the $MnO_2/TiO_2/Ti$ electrode exhibited a larger CV integration area, indicating that the $MnO_2/TiO_2/Ti$ nanocomposites had a higher area capacitance, showing their superior electrochemical properties and the high capacitance behavior of $MnO_2$.

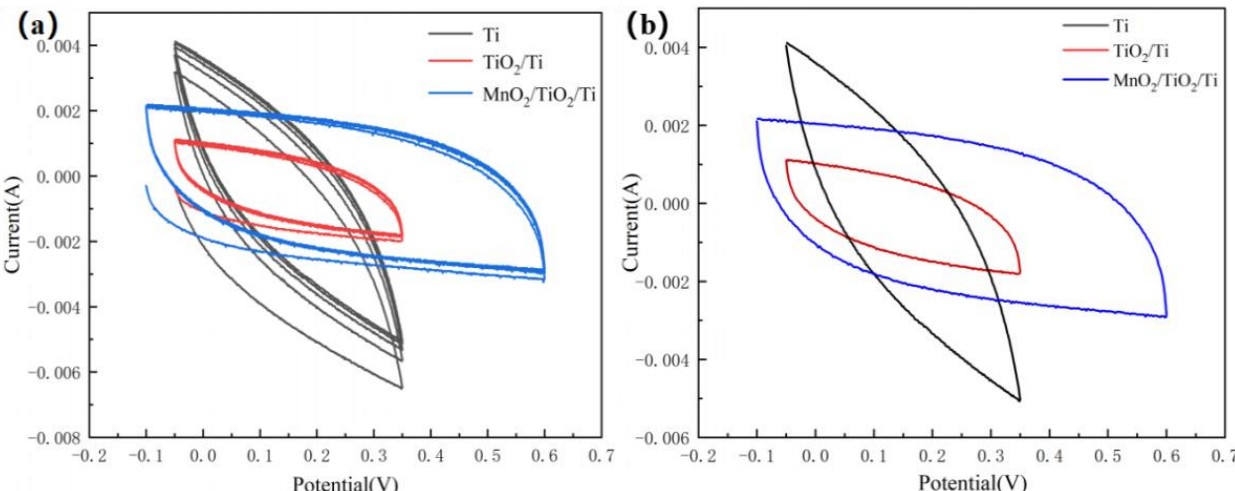

**Figure 6.** (**a**) Cyclic voltammetry of commercialized metal porous Ti, surface-modified TiO$_2$/Ti, and MnO$_2$/TiO$_2$/Ti nanocomposites. (**b**) Graph of the test results for one cycle.

The quality of the capacitance performance of the electrode material can be seen from the closed area of the curve. Titanium dioxide was intended to improve the manganese dioxide loading, although the presence of titanium dioxide did not contribute to the capacitance of the final electrode material; only the loaded manganese dioxide contributed to the capacitance. The area of the closed curve of the MnO$_2$/TiO$_2$/Ti nanocomposite, as shown in Figure 6b, changed greatly with the addition of MnO$_2$, producing the largest closed curve. The specific capacitance (C$_p$) of the porous Ti, surface-modified TiO$_2$/Ti, and MnO$_2$/TiO$_2$/Ti nanocomposites can be estimated using Equation (1):

$$C_P = \frac{A}{2mk(V_2 - V_1)} \tag{1}$$

where C$_p$ is the specific capacitance (F/g), A is the CV curve area (V), m is the mass of the active substance (g), k is the scanning rate of the CV curve (V/s), and (V$_2$ − V$_1$) is the scanning potential range of the CV curve test. The histogram of the Ti, TiO$_2$/Ti, and MnO$_2$/TiO$_2$/Ti specific capacitance values is shown in Figure 7, and the specific capacitance values of the above three samples were obtained according to the CV curve area of Figure 6b, which were 181.9F/g, 88.9F/g, and 314F/g, respectively. The specific capacitance value of the MnO$_2$/TiO$_2$/Ti composite electrode material was significantly higher than that of both of the former, which was 3.5 times larger than that of the TiO$_2$/Ti electrode material. Therefore, loading MnO$_2$ to the TiO$_2$/Ti electrode material had better specific capacitance behavior than the TiO$_2$/Ti and pure Ti electrode materials. The CV results showed that the MnO$_2$/TiO$_2$/Ti nanocomposite resistance was small, and the capacitance performance was improved by the additional MnO$_2$ nano-coating.

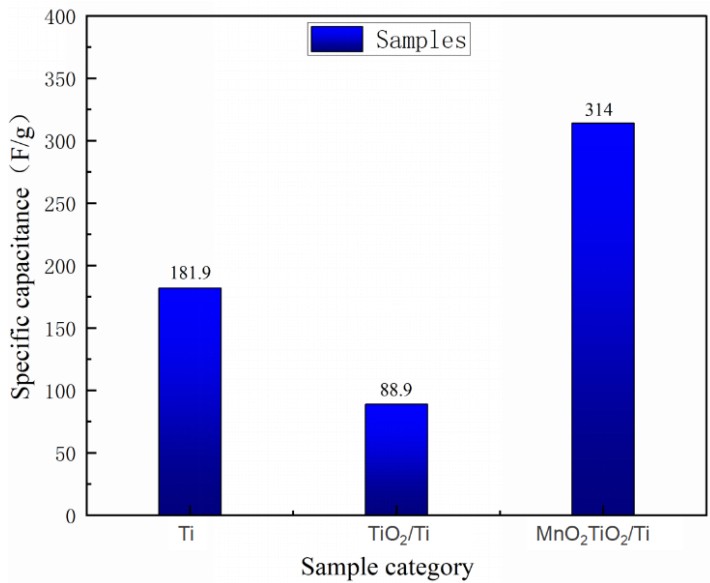

**Figure 7.** Histogram of specific capacitance values of metal porous Ti, surface-modified $TiO_2$/Ti, and $MnO_2$/$TiO_2$/Ti nano composites.

## 4. Conclusions

In summary, a compact $MnO_2$/$TiO_2$ coating on metallic porous Ti was prepared successfully through an alternating immersion method of a chemical precipitation reaction of $KMnO_4$ and $MnSO_4$ aqueous solutions. The morphologies of the coatings and chemical compositions were studied by XRD, Raman, and SEM. The electrochemical behavior of the $MnO_2$/$TiO_2$/Ti nanocomposites was studied via CV, Tafel curves, and EIS. The results revealed that the $MnO_2$/$TiO_2$/Ti exhibited better electrochemical properties and significantly improved the corrosion resistance than those of the pure porous Ti and the surface-modified Ti. In addition, the capacitance performance was improved strongly, and the specific capacitance value reached 314 F/g. $MnO_2$/$TiO_2$/Ti nanocomposites are therefore a promising composite electrode material for supercapacitor applications.

**Author Contributions:** Conceptualization, S.C.; methodology, J.P. and Q.L.; software, X.W.; validation, X.W.; formal analysis, X.W.; investigation, J.P., Q.L. and X.D.; resources, S.C.; data curation, X.W.; writing—original draft preparation, X.W.; writing—review and editing, S.C.; visualization, L.S.; supervision, S.C.; project administration, L.S. All authors have read and agreed to the published version of the manuscript.

**Funding:** This research was financially supported by the 2018 Doctoral Research Funds of Shandong Jianzhu University (X18064Z), Joint Fund Project for Natural Science Foundation of Shandong Province (ZR2021LLZ013).

**Institutional Review Board Statement:** Not applicable.

**Informed Consent Statement:** Not applicable.

**Data Availability Statement:** Not applicable.

**Conflicts of Interest:** The authors declare no conflict of interest.

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
