# Peer review of "Preparation, Corrosion Resistance, and Electrochemical Properties of MnO2/TiO2 Coating on Porous Titanium"

_coatings, doi:10.3390/coatings12101381_

Round 1
Reviewer 1 Report
The authors need to address the following concerns when submitting the revised version.
1. In the Abstract: the authors need to improve with more specific short numeric results and conclusions.
2. The novelty, significance, and importance of this study were not well discussed and illustrated. The objectives of this study should be clearly illustrated to present the significance and importance of the work. The originality of the paper also needs to be further clarified herein.
3. How was the data chosen? How many times every experiment has been repeated?
4. Please add the values of Tafel slopes, Ecorr and Icorr in a Table form. Presenting Tafel slopes is necessary.
5. Please give chi-squared values for EIS results. The fitting parameters should be added to a Table with a proper discussion. Nyquist plots must have equal-scaled axes. The frequency range during EIS measurements should be added to the experimental part.
6. The performance and yielding must be compared with other materials in the literature.
7. Better discussion/data must be demonstrated for the protection mechanism.
8. Authors could refer to the following reports to improve the quality of this work
https://doi.org/10.1016/j.jallcom.2020.156840
https://doi.org/10.3390/met11081182
https://doi.org/10.1016/j.surfcoat.2019.125027
https://doi.org/10.1016/j.corsci.2021.109764
Author Response
Dear Expert,
Thank you very much for giving me a very valuable opportunity to revise my manuscript, and we are very grateful to the editors and reviewers for their positive and constructive comments on our manuscript "Preparation, corrosion resistance, and electrochemical properties of MnO2/TiO2 coating on porous titanium".(Manuscript number:1904317)
We have carefully studied the opinions of experts and editors, marked all the revision points in the paper in red letters, and we have tried our best to improve the manuscript according to the review opinions. The specific reply is as follows:
Expert Review Comments 1:
- In the Abstract: the authors need to improve with more specific short numeric results and conclusions.
【Reply】 Accept the review opinion. Amend as follows:
In this work, MnO2/TiO2 coating on metallic porous titanium was prepared through a hydrothermal-based chemical method, followed by a chemical precipitation reaction of KMnO4 and MnSO4 aqueous solutions. The surface of the MnO2/TiO2/Ti was uniform and compact, with a high load capacity. The corrosion resistance and electrochemical properties of the MnO2/TiO2 coating were investigated in comparison with those of pure Ti and TiO2 coatings. Cyclic voltammetry and constant current charge–discharge measurements showed that the MnO2/TiO2/Ti electrode presented good electrochemical performance. The MnO2/TiO2/Ti electrode had the highest capacitor performance compared to the other electrodes, and the nano-MnO2 coating significantly decreased the corrosion current densities. The nano-MnO2 coating exhibited excellent anti-corrosion properties at room temperature and better capacitance performance compared with pure Ti and TiO2 coatings. After surface modification, TiO2/Ti coated MnO2 had better electrochemical behavior and significantly improved corrosion resistance than the TiO2/Ti nanocomposites. Its specific capacitance reached 314 F/g, which was 3.5 times that of the TiO2/Ti electrode material.
And the manuscript has been revised.
[2]. The novelty, significance, and importance of this study were not well discussed and illustrated. The objectives of this study should be clearly illustrated to present the significance and importance of the work. The originality of the paper also needs to be further clarified herein.
【Reply】Thanks very much for the reviewer’s suggestion. We have already accepted the suggestion. Amend as follows:
“Furthermore, Kaseem and co-workers reported the importance of TiO2 coating in electrochemical and biomedical applications based on plasma electrolytic oxidation or micro-arc oxidation coating treatment.” in the revised introduction.
[3] How was the data chosen? How many times every experiment has been repeated?
【Reply】 Each experiment had been repeated many times, at least 7-10 times, and the selected data can be repeated.
[4] Please add the values of Tafel slopes, Ecorr and Icorr in a Table form. Presenting Tafel slopes is necessary.
【Reply】 Accept the review opinion. Amend as follows:
Table 1. Results of the Tafel texts of the porous Ti, surface-modified TiO2/Ti, and MnO2/TiO2/Ti nanocomposites.
|
Sample |
Tafel polarization |
|||
|
Icorr(uA) |
Ecorr (mV) |
Cathodic Tafel Slope (1/V) |
Anodic Tafel Slope (1/V) |
|
|
Ti |
36.08 |
-320 |
8.221 |
3.019 |
|
TiO2/Ti |
23.93 |
-260 |
9.633 |
2.873 |
|
MnO2/TiO2/Ti |
80.41 |
-100 |
8.224 |
3.091 |
[5] Please give chi-squared values for EIS results. The fitting parameters should be added to a Table with a proper discussion. Nyquist plots must have equal-scaled axes. The frequency range during EIS measurements should be added to the experimental part.
【Reply】 Accept the review opinion. Amend as follows:
Table 2. Results of the EIS fitting texts of the porous Ti, surface-modified TiO2/Ti, and MnO2/TiO2/Ti nano composites.
|
Sample Rs(Ωcm2) Rct(Ωcm2) CPE1-T CPE-P Rct dl(Ωcm2) CPE2-T CPE2-P |
|
Ti 98.02 92.11 2.7x10-3 0.453 194.4 1.1x10-3 1.10 |
|
TiO2/Ti 64.99 37.94 3.0x10-4 0.580 986.1 2.2x10-4 0.74 |
|
MnO2/TiO2/Ti 15.34 4.20 3.17x10-5 0.853 691.7 3.5x10-4 0.72 |
[6] The performance and yielding must be compared with other materials in the literature.
【Reply】Thanks very much for the reviewer’s suggestion. We have already accepted the suggestion. Amend as follows:
“In addition, the electrochemical performance of MnO2 is limited by the electronic conductivity of densely packed matter without a porous structure [30].” In the introduction of the revision.
[7] Better discussion/data must be demonstrated for the protection mechanism.
【Reply】Thanks very much for the reviewer’s suggestion. We have already accepted the suggestion and cited other literature, Amend as follows:
Line 188-191: “Xavier et al. verified that the incorporation of MnO2 nanoparticles reduced the corrosion rate [40]. The presence of MnO2 nanoparticles helps to clog pores and cracks in the coating, hindering the initiation of corrosion at the coating/metal interface.”
[8] Authors could refer to the following reports to improve the quality of this work
https://doi.org/10.1016/j.jallcom.2020.156840
https://doi.org/10.3390/met11081182
https://doi.org/10.1016/j.surfcoat.2019.125027
https://doi.org/10.1016/j.corsci.2021.109764
【Reply】Thanks very much for the reviewer’s suggestion. We have already accepted the suggestion. We have studied the above literatures and cited them in the revision [26-29].
Reviewer 2 Report
The summary can be worked on in order to identify in detail the novelty and relevance of the study. The various studies carried out are indicated, which allowed to obtain materials with particular qualities (possible applications). It is not described whether any novel or frontier methodology is presented for the design of such materials. These are the relevance of the proposed study. It is recommended to work in this direction, to increase the impact.
The state of the art can be expanded with literature of impact on the fear under study. Regarding the relevance, it could be appropriate oriented to what is indicated in the document, which indicates the need to develop new materials for possible application in energy storage systems. Conclusions could still go in this direction. Remembering that it is a suggestion.
The methodology is described appropriately.
Line 137: Because it is necessary to determine the presence of MnO2.
Line 146-151: Relevance of MnO2 presence is resumed. it is indicated that you could have an MnO2/TiO2/Ti nanocomposite, perhaps further discussion/analysis with the appropriate scientific support is required. On the respective relevance of this compound. That seems to play a preponderant role in the study. And perhaps here goes the novelty or relevance of the study.
Lines 155-174: What is the relevance of knowing morphology: porosity, amorphous material, nanomaterials are mentioned (but SEM has the ability to see these sizes?). Improve the discussion, perhaps support with DRX or Raman for possible crystals of TiO2 and other possible types of crystals that were asked to be present in the material. But describe the importance of such morphology. Not just describe in a general way. Some statements, it is advisable to indicate supporting scientific reference.
Lines 178-176: The result obtained is interesting, and again MnO2 plays an important role. This paragraph describes or concludes that the composite material impacts the electrode dissolution process and improved anticorrosive performance. But there is no in-depth analysis of why this compound impacts on the properties of interest, I think it is necessary to respond to this explanation of why the polarization curves are affected by having this compound.
Electrochemical study of impedance and capacitance, another ves show an important difference by the presence of MnO2, but the why is not described.
The conclusions are general, it is recommended to improve, which could be referred to why MnO2 allows a better electrochemical behavior and corrosion resistance, with respect to materials that do not contain this compound. Because MnO2 offers a synergy in the properties of interest.
It is recommended to update state of the art, current scientific literature.
Author Response
Dear Expert,
Thank you very much for giving me a very valuable opportunity to revise my manuscript, and we are very grateful to the editors and reviewers for their positive and constructive comments on our manuscript "Preparation, corrosion resistance, and electrochemical properties of MnO2/TiO2 coating on porous titanium".(Manuscript number:1904317)
We have carefully studied the opinions of experts and editors, marked all the revision points in the paper in red letters, and we have tried our best to improve the manuscript according to the review opinions. The specific reply is as follows:
Expert Review Comments 2:
The summary can be worked on in order to identify in detail the novelty and relevance of the study. The various studies carried out are indicated, which allowed to obtain materials with particular qualities (possible applications). It is not described whether any novel or frontier methodology is presented for the design of such materials. These are the relevance of the proposed study. It is recommended to work in this direction, to increase the impact.
【Reply】 Thanks to the valuable suggestions made by the editor and expert, we accept them very sincerely and have made corresponding additions and modifications in the text.
The state of the art can be expanded with literature of impact on the fear under study. Regarding the relevance, it could be appropriate oriented to what is indicated in the document, which indicates the need to develop new materials for possible application in energy storage systems. Conclusions could still go in this direction. Remembering that it is a suggestion.
【Reply】 Thanks to the valuable suggestions made by the editors and experts, we accept them very sincerely and will continue to improve in the follow-up research based on the suggestions made by the reviewers and experts.
Line 137: Because it is necessary to determine the presence of MnO2.
【Reply】Thanks to the valuable advice of the editors and experts, the corresponding line 137 is an XRD test of the sample produced. The presence of MnO2 was not found in the XRD atlas, so the presence of manganese dioxide could not be verified, and the presence of manganese dioxide was confirmed in further Raman tests.
Line 146-151: Relevance of MnO2 presence is resumed. it is indicated that you could have an MnO2/TiO2/Ti nanocomposite, perhaps further discussion/analysis with the appropriate scientific support is required. On the respective relevance of this compound. That seems to play a preponderant role in the study. And perhaps here goes the novelty or relevance of the study.
【Reply】 Thanks to the valuable suggestions made by the editors and experts, we accept them very sincerely and will continue to improve in the follow-up research based on the suggestions made by the reviewers and experts.
Lines 155-174: What is the relevance of knowing morphology: porosity, amorphous material, nanomaterials are mentioned (but SEM has the ability to see these sizes?). Improve the discussion, perhaps support with DRX or Raman for possible crystals of TiO2 and other possible types of crystals that were asked to be present in the material. But describe the importance of such morphology. Not just describe in a general way. Some statements, it is advisable to indicate supporting scientific reference.
【Reply】Thanks very much for the reviewer’s suggestion. We have already accepted the suggestion. This material is based on the surface of the porous metallurgical titanium substrate. There are many pores inside it. After the surface is oxidized to TiO2 and then loaded with MnO2 nanoparticles, the electron transport channel will become longer.【Y.X. Zhang, M. Kuang, X.D. Hao, Y. Liu, M. Huang, X.L. Guo, J. Yan, G.Q. Han, and J. Li, Rational Design of Hierarchically Porous Birnessite-Type Manganese Dioxides Nanosheets on Dif-ferent One-Dimensional Titania-Based Nanowires for High Per-formance Supercapacitors. J. Power Sources 270, 675 (2014).】【J.-G. Wang, Y. Yang, Z.-H. Huang, and F. Kang, Coaxial Carbon Nanofibers/MnO2 Nanocomposites as Freestanding Electrodes for High-Performance Electrochemical Capacitors. Electrochim. Acta 56, 9240 (2011).】
Lines 178-176: The result obtained is interesting, and again MnO2 plays an important role. This paragraph describes or concludes that the composite material impacts the electrode dissolution process and improved anticorrosive performance. But there is no in-depth analysis of why this compound impacts on the properties of interest, I think it is necessary to respond to this explanation of why the polarization curves are affected by having this compound.
【Reply】Thanks very much for the reviewer’s suggestion. We have already accepted the suggestion. The modified part has been highlighted in the revision.
Electrochemical study of impedance and capacitance, another ves show an important difference by the presence of MnO2, but the why is not described.
The conclusions are general, it is recommended to improve, which could be referred to why MnO2 allows a better electrochemical behavior and corrosion resistance, with respect to materials that do not contain this compound. Because MnO2 offers a synergy in the properties of interest.
【Reply】Thanks very much for the reviewer’s suggestion. We have already accepted the suggestion. The incorporation of MnO2 nanoparticles reduces the corrosion rate. The presence of MnO2 nanoparticles helps to clog pores and cracks in the coating, hindering the initiation of corrosion at the coating/metal interface.【Corrosion protection performance and interfacial interactions polythiophene/silanes/MnO2 nanocomposite coatings on magnesium alloy in marine environment】On this basis, we have also made corresponding additions and modifications in the article
It is recommended to update state of the art, current scientific literature.
【Reply】Thanks very much for the reviewer’s suggestion. We have already accepted the suggestion. Based on the advice of editors and experts, we have cited the latest literature in the article and revised it accordingly.
Reviewer 3 Report
On my opinion, the samples could be characterized more completely. Particularly, it would be reasonable to explain the necessary of six times precipitation of MnO2, magic 30 s for all procedures including the washing with deionized water and then soaking in water for 15 min. The justifications of all these numbers, e.g., SEM (although SEM (Fig. 3) images give only external surface morphology. To say exactly what a substance is located on the external surface (e-f) must be presented additional data), quantity of MnO2 deposited, electrochemical characteristics and so on after each procedure should be presented. What concretely does occur with the sample after any procedures?
MnO2 was obtained by reaction KMnO4 with MnSO4 in water (it is not “co-precipitation” (lines 11, 62). But it is well known that in this case, MnO2×xH2O (or MnO(OH)2) is formed. To obtain MnO2 annealing is needed. To justified MnO2 formation the data presented are absolutely insufficient.
Rather molar capacity describes more adequately the advantages of TiO2/Ti over MnO2/TiO2/Ti samples. It should be presented the calculations of specific capacity.
The improvement of anti-corrosion resistance after precipitating the semi-conductor on the metal is a quite trivial result. It is of interest to understand how namely MnO2 provides this feature.
It should be also explained how TiO2 does improve MnO2 loading (lines 224-225). In general, it is unclear what quantity of metal oxides (TiO2 and MnO2) was loaded.
Where is “a small red-ox peak” (line 215)?
“…the presence of titanium dioxide does not contribute to the capacitance of the final electrode material…” (lines 225-226) - it should be commended.
Terminology: Line 169 …KMnO4 and MnSO4 were uniformly deposited on TiO2/Ti formed by a disproportionation reaction…and Figure 3e-f is consistent with the findings above …it is unclear and unsubstantiated! Reaction KMnO4 and MnSO4 is the reaction of co-proportionation rather than disproportionation.
Lines 31,36 (EDLs), 48 (ferric trioxide should be diferric trioxide), 11 (co-precipitation reaction should be precipitation).
Author Response
On my opinion, the samples could be characterized more completely. Particularly, it would be reasonable to explain the necessary of six times precipitation of MnO2, magic 30 s for all procedures including the washing with deionized water and then soaking in water for 15 min. The justifications of all these numbers, e.g., SEM (although SEM (Fig. 3) images give only external surface morphology. To say exactly what a substance is located on the external surface (e-f) must be presented additional data), quantity of MnO2 deposited, electrochemical characteristics and so on after each procedure should be presented. What concretely does occur with the sample after any procedures?
MnO2 was obtained by reaction KMnO4 with MnSO4 in water (it is not “co-precipitation” (lines 11, 62). But it is well known that in this case, MnO2×xH2O (or MnO(OH)2) is formed. To obtain MnO2 annealing is needed. To justified MnO2 formation the data presented are absolutely insufficient.
Rather molar capacity describes more adequately the advantages of TiO2/Ti over MnO2/TiO2/Ti samples. It should be presented the calculations of specific capacity.
【Response】Thanks very much for the reviewer’s suggestion. We have already accepted the suggestion.
The presence of MnO2 was not found in the XRD atlas, so the presence of manganese dioxide could not be verified, and the presence of manganese dioxide was confirmed in further Raman tests.
Addition, amend as follows:
(1)
where is the specific capacitance (F/g), A is the CV curve area (V), m is the mass of the active substance (g), k is the scanning rate of the CV curve (V/s), and is the scanning potential range of the CV curve test.
The improvement of anti-corrosion resistance after precipitating the semi-conductor on the metal is a quite trivial result. It is of interest to understand how namely MnO2 provides this feature.
It should be also explained how TiO2 does improve MnO2 loading (lines 224-225). In general, it is unclear what quantity of metal oxides (TiO2 and MnO2) was loaded.
Where is “a small red-ox peak” (line 215)?
“…the presence of titanium dioxide does not contribute to the capacitance of the final electrode material…” (lines 225-226) - it should be commended.
【Response】Thanks very much for the reviewer’s suggestion. We have already accepted the suggestion.
Lines 257-260:“Titanium dioxide was intended to improve manganese dioxide loading, although the presence of titanium dioxide does not contribute to the capacitance of the final electrode material; only the loaded manganese dioxide contributes to the capacitance.”The electrochemical results of nano-composites are repeatable.
Line 251 :“a small red-ox peak” was cited by the literature [Ramadoss, A.; Kim, S.J. Hierarchically structured TiO2-MnO2 nanowall arrays as potential electrode material for high performance supercapacitors. Int J Hydrogen Energy 2014, 39, 12201-12212.]
MnO2 and TiO2 have a very good synergistic effect. Line 188-191: Xavier et al. verified that the incorporation of MnO2 nanoparticles reduced the corrosion rate [40]. The presence of MnO2 nanoparticles helps to clog pores and cracks in the coating, hindering the initiation of corrosion at the coating/metal interface.
The revision has been modified, and the cited literature is already in the revised version.
Terminology: Line 169 …KMnO4 and MnSO4 were uniformly deposited on TiO2/Ti formed by a disproportionation reaction…and Figure 3e-f is consistent with the findings above …it is unclear and unsubstantiated! Reaction KMnO4 and MnSO4 is the reaction of co-proportionation rather than disproportionation.
【Response】Thanks very much for the reviewer’s suggestion. We have already accepted the suggestion.
Lines 174-175: Here, KMnO4 and MnSO4 were uniformly deposited on TiO2/Ti formed by a co-proportionation reaction.
Lines 31,36 (EDLs), 48 (ferric trioxide should be diferric trioxide), 11 (co-precipitation reaction should be precipitation).
【Response】Thanks very much for the reviewer’s suggestion. We have already accepted the suggestion.
The modified part has been highlighted in the revision.
Round 2
Reviewer 1 Report
1. The unit of Tafel slopes should V/decade
2. Cathodic Tafel slope should be negative
3. Frequencies should be added to the Nyquist plots. Otherwise, Bode plots should be inserted. ( This comment was not answered yet in the revised version)
Author Response
- The unit of Tafel slopes should V/decade
Response 1: Thanks to the valuable suggestion made by the reviewer, we accept them very sincerely and have made corresponding modifications in the revised version.
- Cathodic Tafel slope should be negative
Response 2: Thanks to the valuable suggestion made by the reviewer, we accept them very sincerely and have made corresponding modifications in the revision.
- Frequencies should be added to the Nyquist plots. Otherwise, Bode plots should be inserted. (This comment was not answered yet in the revised version)
Response 3: Thank you very much for the reviewer's suggestion, we have accepted this suggestion and attached a picture in the review comments, once again thank the reviewer for the valuable suggestions, so that our article is more substantial. The bode diagram is shown in the figure. The results obtained by the Bode plots are consistent with the Nyquist diagram.

Reviewer 2 Report
The publication of the article is recommended, after the authors improved the discussion in the requested points of interest. They describe and respond to the various external doubts. The conclusions could be improved. An adequate analysis of the state of the art is presented, which supports the discussion of results. The relevance is clarified.
Author Response
Response : Thanks to the valuable suggestion made by the reviewer, we accept them very sincerely and have made corresponding modifications in the revised version.

Reviewer 3 Report
Thank you, it is all right.
Author Response
Thanks to the valuable suggestion made by the reviewer